# Extracellular Targets to Reduce Excessive Scarring in Response to Tissue Injury

**DOI:** 10.3390/biom13050758

**Published:** 2023-04-27

**Authors:** Jolanta Fertala, Mark L. Wang, Michael Rivlin, Pedro K. Beredjiklian, Joseph Abboud, William V. Arnold, Andrzej Fertala

**Affiliations:** 1Department of Orthopaedic Surgery, Sidney Kimmel Medical College, Thomas Jefferson University, Philadelphia, PA 19107, USA; 2Rothman Institute of Orthopaedics, Thomas Jefferson University Hospital, Philadelphia, PA 19107, USA

**Keywords:** fibrosis, excessive scarring, extracellular matrix, collagen, anti-fibrotic targets, matrix stiffness, wound healing

## Abstract

Excessive scar formation is a hallmark of localized and systemic fibrotic disorders. Despite extensive studies to define valid anti-fibrotic targets and develop effective therapeutics, progressive fibrosis remains a significant medical problem. Regardless of the injury type or location of wounded tissue, excessive production and accumulation of collagen-rich extracellular matrix is the common denominator of all fibrotic disorders. A long-standing dogma was that anti-fibrotic approaches should focus on overall intracellular processes that drive fibrotic scarring. Because of the poor outcomes of these approaches, scientific efforts now focus on regulating the extracellular components of fibrotic tissues. Crucial extracellular players include cellular receptors of matrix components, macromolecules that form the matrix architecture, auxiliary proteins that facilitate the formation of stiff scar tissue, matricellular proteins, and extracellular vesicles that modulate matrix homeostasis. This review summarizes studies targeting the extracellular aspects of fibrotic tissue synthesis, presents the rationale for these studies, and discusses the progress and limitations of current extracellular approaches to limit fibrotic healing.

## 1. Introduction

The ability to heal injured tissues is fundamental for survival. Natural healing usually includes scar formation, a process accelerated by inflammation. While scars patch the injury sites, excessive scarring alters critical tissue and organ functions. 

Regardless of the injury site, tissue type, and nature of the injury, the healing process includes hemostasis, inflammation, proliferation, and remodeling. For instance, in response to acute or systemic injury, blood-derived and local inflammatory cells migrate to damaged sites and set the stage for tissue repair by producing many growth factors. Subsequently, these growth factors stimulate resident and migratory fibroblastic cells, increasing their proliferation and the biosynthesis of scar tissue materials. Myofibroblasts are crucial producers of scar-building elements, including fibrillar collagens, fibronectin, proteoglycans, glycosaminoglycans, and others (Figure 1) [1,2].

While complete regeneration of injured adult tissues (i.e., returning to their original state) is a rare phenomenon, in some fetal tissues, regeneration may occur [3]. Essential elements for the regeneration of mature tissues include an intact extracellular matrix (ECM) and tissue-specific cells able to synthesize damaged components. For example, hepatocytes can regenerate the liver following acute toxic injury if the ECM is undamaged. Studies have demonstrated that the hepatocytes that perform regeneration are derived from local or circulating stem cells or mature hepatocytes that re-entered the cell cycle [4]. In contrast, chronic or traumatic acute injuries that damage the ECM architecture make healing by regeneration impossible. Consequently, healing by scarring helps to maintain the function and structural integrity of wounded tissues.

Although balanced scar formation maintains tissue integrity, excessive scarring in various tissues and organs is a significant medical problem. According to some estimates, fibrotic disorders are associated with 45% of all deaths [5]. Despite the enormous burden caused by these disorders, attempts to treat them have been largely unsuccessful. Consequently, researchers continue to define anti-fibrotic targets and design relevant inhibitors to block or reverse the fibrotic scarring process. This review focuses on one of these targets, namely the formation of collagen-rich ECM that defines the fibrotic deposits.

## 2. Collagen-Rich Matrix, a Versatile Biological Patch

Although healthy tissues differ in the cell types, molecular composition, and architecture of their ECM, the scar formation steps and fundamental elements of fibrotic deposits are similar.

For example, healthy cartilage mainly consists of collagen II-rich fibrils, pericellular collagen VI, fibril-associated collagen IX, and proteoglycans that form a structure able to withstand compression forces. Cartilage-specific cells, chondrocytes, control cartilage homeostasis and maintain cartilaginous structure [6].

The cellular makeup of the kidney, however, is more complex. It includes smooth muscle cells, podocytes, pericytes, fibroblasts, and many other cell types. Similarly, the pool of collagenous proteins differs and includes collagen I, collagen III, relatively large amounts of collagen IV, collagen VI, and a few additional collagen types [7].

Despite the differences in the cellular and ECM ingredients of healthy cartilage and kidney, both heal by scar formation. In both tissues, the main component of the scar is collagen I-rich fibrils [8,9]. Because similar mechanisms function in other tissues and organs, the formation of collagen I-rich scars represents a ubiquitous tissue repair mechanism, regardless of the wounded tissue’s injury type or location.

Although scar-based repair appears to be nature’s way to fix injuries without needing multiple unique and tissue-specific repair mechanisms, the tradeoff is that, when formed excessively, significant scarring can severely alter the repaired tissue architecture and function.

## 3. Excessive Scar Formation That Alters Vital Functions of Affected Tissues and Organs

Excessive scarring of the skin, tendon, muscle, and ligament alters the mechanical functions of these tissues. Similarly, ocular scars may impair vision, while the excessive formation of fibrotic deposits can alter vocal cords and harm speech. Post-traumatic scarring of peripheral nerves prevents the regeneration of the axons and blocks their growth from the proximal toward the distal stump of an injured nerve [10,11,12,13,14,15].

Similar consequences of excessive formation of collagen I-rich deposits have also been observed in organ fibrosis disorders caused by chronic inflammation, often with no defined injury events. For instance, patients with idiopathic pulmonary fibrosis (IPF) develop collagen I-rich thick scar tissue that blocks an efficient oxygen exchange, ultimately leading to death. In addition, as indicated above, fibrotic deposits in the liver and kidney alter these organs’ proper functions and may lead to their failure [16,17]. Further, patients with scleroderma develop stiff skin and, in later stages of the disease, progress toward fibrosis of multiple sites, including joints, lungs, esophagus, heart, and other organs [18].

These examples illustrate that repairing acute local and systemic injuries is a complex balancing act that can rapidly shift from needed tissue repair to unwanted fibrotic scarring, with severe outcomes.

## 4. Anti-Fibrotic Treatment: A Challenging Task

Even with decades-long studies attempting to define anti-fibrotic treatments, validate anti-fibrotic targets, and produce valuable therapeutics, effective and safe therapies designed to limit excessive scarring have yet to be developed.

Nintedanib and pirfenidone were only recently approved by the Food and Drug Administration (FDA) for patients with IPF. Clinical data indicate that these drugs slow down the rate of decline in forced vital capacity (FVC). However, studies did not show conclusively if these drugs reduce the mortality of patients with IPF [19].

Furthermore, tests of nintedanib applied to reduce pleuroparenchymal fibroelastosis, a subtype of interstitial pneumonia with upper lobe fibrosis, showed limited efficacy of this drug compared with that for the IPF treatment [20]. These studies suggest only a limited utility of nintedanib for treating fibrotic disorders of the lungs.

The anti-fibrotic mechanism of these FDA-approved drugs remains unclear. Some have suggested that they have broad anti-inflammatory effects and reduce the production of pro-fibrotic factors and matrix elements, including transforming growth factor-beta 1 (TGF-β1), tumor necrosis factor-alpha (TNF-α), platelet-derived growth factor (PDGF), interleukin 1 beta (IL-1β), and collagen I [21]. Other studies indicate that the anti-fibrotic mechanisms of these drugs may directly block collagen fibrillogenesis [22].

Nintedanib and pirfenidone were also tested as inhibitors in many other fibrotic conditions in models of excessive skin, eye, and muscle scarring [23,24,25,26]. Although the drugs demonstrated anti-fibrotic properties in some of these tests, they have yet to be applied clinically to treat fibrotic disorders other than IPF.

### 4.1. Targeting Pro-Fibrotic Cells

Myofibroblasts that elaborate fibrosis, intracellular processes that drive excessive scarring, and the extracellular steps of the scar matrix assembly are recognized as anti-fibrotic targets. Strategies to limit the pro-fibrotic behavior of the myofibroblasts include using anti-proliferative agents, blocking the transition of fibroblastic and epithelial cells to myofibroblasts, and inhibiting circulating pro-fibrotic cells from homing in on injury sites [27].

Since many of these processes are controlled by TGF-β1, this cytokine and associated mediators of its activity, including connective tissue growth factor (CTGF), represent crucial anti-fibrotic targets.

Despite the crucial roles of cells, pro-fibrotic intracellular processes, and growth factors associated with excessive scarring, no adequate specific treatments that aim at these targets have been developed for clinical use. Although the reasons for the poor outcomes are unclear, the literature points to several problems hampering the development of successful anti-fibrotic approaches. One problem is the natural redundancy of injury repair mechanisms that utilize multiple pathways to form collagen I fibril-rich scars. In fibrosis, these mechanisms are preserved and active in all tissues. Therefore, targeting only one mechanism or pathway to block the fibrotic process is likely insufficient [28].

### 4.2. Stiff ECM: A Crucial Pro-Fibrotic Culprit

The common denominator of different scarring mechanisms is the end product of the scarring process, namely, collagen I-based fibrotic neotissue. Following initial synthesis, this tissue stiffens, altering crucial natural functions of repaired sites.

Studies have demonstrated that stiff tissue is a crucial pro-fibrotic stimulant of fibroblasts and inflammatory cells in injury sites (Figure 2) [28,29,30].

#### 4.2.1. Fibroblasts

Local fibroblasts that reside in wounded sites and fibroblasts that migrate from distant locations perform crucial tasks in balanced wound healing and fibrotic scarring (Figure 2) [31]. Many growth factors modulate these tasks, with TGF-β1 playing the central role [31,32].

As indicated above, contractile myofibroblasts that express α smooth muscle actin (αSMA) incorporated into stress fibrils are a hallmark of fibrosis [31]. Crucial functions of these cells include wound contracture and the production of elements of the ECM, including collagenous proteins.

Myofibroblasts interact with inflammatory cells, including macrophages and mast cells. These cells influence fibroblast activities by secreting TGF-β1, PDGF, vascular endothelial growth factor (VEGF), IL-6, and IL-13. In turn, fibroblasts impact the macrophages’ phenotype and function by changing the physical properties of the ECM [31].

In wound healing without excessive fibrosis, myofibroblasts ultimately cease their functions. They may revert to an “inactive fibroblast” state, enter senescence, or be eliminated via apoptosis at the end of a routine healing process. In contrast, in excessive scarring, myofibroblasts remain active in the fibrotic processes, resisting apoptosis while continuing their pro-fibrotic activities [33].

#### 4.2.2. Inflammatory Cells

Inflammatory cells that drive fibrotic healing include mast cells and macrophages (Figure 2) [31]. Studies have demonstrated that the stiff ECM environment enhances the pro-fibrotic behavior of mast cells and promotes their durotaxis, i.e., migration along stiffness gradients (Figure 2) [34]. Consequently, Hildebrand et al. targeted these cells with ketotifen to reduce the progress of fibrotic healing after an elbow injury. However, clinical trials in Canada demonstrated that this approach did not significantly mitigate post-traumatic elbow stiffness [35].

These studies indicate that aiming at mast cells alone is insufficient to reduce fibrotic healing in a clinically relevant way.

Macrophages show stiffness-dependent pro-fibrotic behavior too [36]. As previously demonstrated, mechano-gated ion channels and the α2β1 integrin, a member of the integrin family of heterodimeric cellular receptors, mediate the mechanical activation of these cells via complex signaling pathways that promote cell migration, proliferation, and ECM production [31,37].

## 5. Targeting the ECM Stiffness

Considering the crucial pro-fibrotic properties of scar neotissue, some have suggested that targeting the stiff ECM formation and modulating its pro-fibrotic signals may be a game-changer for developing anti-fibrotic therapies [28].

Here, we focus on crucial processes and factors that increase ECM stiffness and contribute to pro-fibrotic mechanotransduction. They include (i) extracellular regulators of scar production, e.g., TGF-β1; (ii) extracellular ECM assembly, e.g., collagen fibrillogenesis; and (iii) cellular receptors that allow the ECM–cell communication, specifically integrins [28,29,30,38].

### 5.1. Production of a Crucial Precursor of Fibrotic Deposits

Collagen I is the main component of scars formed in the skin, musculoskeletal systems, peripheral nerves, the eye, abdomen, spinal cord, and others [39,40,41,42,43,44]. This protein constitutes the most considerable portion of fibrotic tissues formed due to injury in the internal organs, including the liver, lungs, heart, and kidney [45].

The fibrillar architecture formed by this collagen type provides mechanical stability to the injury sites. Nevertheless, collagen I-rich deposits are the main factor causing harmful consequences associated with excessive scarring. These fibrillar structures are produced in a complex process that includes intracellular and extracellular steps (Figure 3).

#### 5.1.1. Intracellular Steps of Collagen I Synthesis

Each collagen molecule comprises three collagen α-chains, that associate in the endoplasmic reticulum (ER) into a triple-helical structure. In the fibril-forming collagens, including collagen I, each chain consists of approximately 330 uninterrupted repeats of -G-X-Y- triplets, in which the -X- and -Y- positions are frequently occupied by proline residues [46].

The fibril-forming collagens are produced as procollagens, in which the triple-helical domains are flanked by globular N-terminal and C-terminal propeptides (Figure 3). Relatively short non-triple-helical telopeptides separate the propeptides and the triple-helical domain.

Post-translational modifications of nascent procollagen α-chains are vital steps that determine collagen molecules’ proper thermostability and mechanical properties. In particular, proline and lysine residues present in the -Y- positions of the -G-X-Y- triplets are hydroxylated by prolyl 4-hydroxylase (P4H) and lysyl hydroxylase (LH), respectively [46].

P4H is a tetramer formed by two catalytic α subunits (P4Hα) and two non-catalytic β subunits (P4Hβ). P4Hβ also serves as protein disulfide isomerase (PDI) and a protein chaperone that prevents premature aggregation of procollagen chains [47,48]. 3-Hydroxyproline residues are also present in the -X- and -Y- positions of the –G-X-Y- triplets [49]. In procollagen I, only one proline residue of the α1(I) chain is 3-hydroxylated [50].

Mature procollagen chains assemble into a triple-helical conformation by a zipper-like folding mechanism [51]. Specialized chaperone proteins stabilize the procollagen molecules and prevent their aggregation. Chaperones involved in procollagen biosynthesis include (i) heat-shock protein 47 (HSP47), (ii) heat-shock 70 kDa-related luminal binding protein (BiP), and (iii) P4Hβ/PDI [52].

#### 5.1.2. Extracellular Procollagen I Processing

Following secretion to the extracellular space, enzymatic cleavage of procollagen propeptides triggers collagen fibril formation [53]. A group of proteolytic enzymes, including a disintegrin and metalloprotease with thrombospondin motifs (ADAMTS)-2, -3, and -14, cleaves the N-terminal propeptides [54]. Among these enzymes, ADAMTS-2, or procollagen N proteinase (PNP), is abundant in collagen I-rich tissues, including skin, tendon, bone, eye, and others [55].

Another group of enzymes, from the tolloid family of zinc metalloproteinases, cleaves the C-terminal propeptides of fibrillar procollagens. Among these metalloproteases, procollagen C proteinase (PCP), also known as bone morphogenetic protein-1 (BMP-1), plays a pivotal role [56].

Further, PCP enhancer (PCPE) participates in the cleavage of procollagen I C propeptides by increasing the rate of the propeptide cleavage up to 20-fold [57]. Studies have shown that this protein is upregulated in many fibrotic conditions, including hypertrophic, keloid, and ocular scars. The increased production of PCPE was also demonstrated in organ fibrosis models [58,59].

Researchers have demonstrated that other enzymes might also process procollagen propeptides. These enzymes include meprins and mast cell chymase, whose activity increases during inflammation and fibrosis [60,61].

#### 5.1.3. Extracellular Assembly of Collagen Fibrils

Following the cleavage of the procollagen propeptides, collagen I molecules self-assemble to form fibrils in a process driven by site-specific interactions among individual collagen molecules [62]. The binding interaction between the C-terminal telopeptides of one collagen molecule and an interacting partner’s telopeptide-binding region (TBR) facilitates crucial nucleation and a proper staggered alignment of collagen molecules [63,64,65]. This interaction is ground zero for the collagen fibril growth in physiological conditions and during the formation of scar deposits (Figure 3).

#### 5.1.4. Cross-Linking of Collagen Fibrils and its Impact on ECM Stiffness

The assembly of individual collagen I molecules into fibrils is an entropy-driven process [53]. The nascent fibrils are held together by electrostatic and hydrophobic forces. However, these fibrils are unstable and may dissociate back into collagen molecules by changing optimal temperature or solvent conditions [66,67].

Thus, the assembled fibrils must be stabilized by covalent bonds between the individual collagen molecules that build them. Ultimately, these bonds, or cross-links, define the resistance of collagen fibrils to proteolytic degradation and determine their mechanical strength (Figure 3).

The hydroxylation of selected lysine residues, catalyzed by the lysyl oxidase (LOX) family of enzymes, facilitates the formation of collagen cross-links [48]. Transglutaminase 2 (TG2) also catalyzes the formation of fibril-stabilizing cross-links [68,69,70].

Collagen fibril formation is a prerequisite for the formation of the cross-links. The collagen molecules must first be arranged in the staggered, D-periodic pattern to allow the LOX enzymes to catalyze the cross-linking reaction.

#### 5.1.5. Collagen Fibrillogenesis: A Crucial Anti-Fibrotic Target

Ultimately, the number of collagen fibrils, their spatial organization, and the extent of their cross-linking define the stiffness of the scar tissue and impact the severity of fibrotic disorders. Therefore, reducing the number of fibrils is the goal of all anti-fibrotic approaches, regardless of whether they aim at cells, intracellular, or extracellular targets.

## 6. Mediators of the Stiffness-Dependent Signals

The stiff-ECM-derived mechanical signals upregulate the expression of macromolecules that build the fibrotic scars. The mechanotransduction that modulates this expression involves integrins. These receptors facilitate cell signaling by forming focal adhesion structures that include talin, vinculin, focal adhesion kinase (FAK), and paxillin (Figure 4).

DDRs also participate in pro-fibrotic mechanotransduction [71,72]. Research data indicate that DDR1 regulates cell adhesion and migration through collagen-rich matrices by associating with non-muscle myosin IIA [73]. This protein is a hexameric enzyme with ATPase activity. It can bind to the actin cytoskeleton, generate forces that shape cell architecture, and facilitate cell motility (Figure 4) [74,75].

Key mediators of fibrotic processes are TGF-β receptors (TGFBR), which regulate the ECM–cell signaling via binding its TGF-β ligands and modulate SMAD-mediated processes (Figure 4). As reviewed by Abuammah et al., under low-shear mechanical conditions, the TGF-β1-dependent SMAD-2 signaling is upregulated [76]. The authors suggested that this mechanism may increase epithelial-to-mesenchymal transition, increasing fibrotic responses in some tissues [77].

Additional mechanotransduction pathways depend on regulating mechanosensitive ion channels via the tether force mechanism facilitated by the extracellular and intracellular partners [38,78].

The nucleus also participates in mechanosensing [79]. One of the elements facilitating the mechanosensing functions of the nucleus is lamin-A, whose production correlates positively with collagen-dependent ECM stiffness [80]. Other central players include yes-associated protein (YAP) and transcriptional co-activator with PDZ-binding motif (TAZ). Under low mechanical loading, they are located in the cytoplasm, where proteasomes degrade them. In contrast, YAP and TAZ escape proteasomal degradation in a high-mechanical-loading environment and translocate to the nucleus. There, they upregulate fibroblasts’ activities, including proliferation, differentiation, suppression of apoptosis, and matrix production [81,82,83,84]. Studies have also demonstrated that in addition to YAP/TAZ signaling, myocardin-related transcription factor (MRTF) participates in ECM stiffness-dependent fibroblast activities (Figure 4) [85,86].

## 7. Mechanotherapeutics

Because of the central role of the factors that mediate the stiffness-dependent pro-fibrotic cell behavior, targeting them is an attractive approach to reduce fibrotic scarring (Figure 4). Efforts to block pro-fibrotic cells and some canonical intracellular mediators of fibrotic healing, however, have been unsuccessful [29]. Significant concerns that hamper the bench-to-bed transition of these efforts include poor specificity of the blockers and unwanted side effects [28].

Targets for the mechanotherapeutics tested thus far are mainly downstream of the central pro-fibrotic physical stimulant, namely the stiff ECM (Figure 3). Aiming at less explored upstream targets directly associated with steps stiffening the matrix, e.g., collagen fibrillogenesis, offers an attractive yet poorly exploited alternative for reducing excessive fibrosis (Figure 3).

### 7.1. Targeting Procollagen Processing

As indicated above, one of the necessary conditions for forming proper collagen fibrils is extracellular, enzymatic removal of procollagen N-terminal and C-terminal propeptides (Figure 3). Studies have demonstrated that the presence of both propeptides precludes the formation of functional stable fibrils. Even the presence of only one of the propeptides leads to the assembly of abnormal tape-like or sheet-like structures [87]. In dermatosporaxis, where the N propeptides have not been removed from the procollagen I molecules, similar structures weaken the architecture of collagen I-rich tissues, most notably the skin [88].

Because of the importance of procollagen I propeptide cleavage for fibril formation, PNP and PCP were identified as attractive anti-fibrotic targets. The rationale for targeting these enzymes was that inhibiting their activities would prevent the cleavage of procollagen propeptides, thereby preventing collagen fibril formation and limiting excessive scarring (Figure 3).

In one study, Ovens et al. synthesized acidic dipeptide hydroxamate inhibitors of PCP. They demonstrated their utility in inhibiting PCP in vitro [89]. Although inhibitors of PNP were reported, they showed broad inhibitory properties, limiting their potential use as specific inhibitors of the N propeptide cleavage [90].

Mouse-based experiments, in which scientists knocked out genes encoding PCP or PNP, demonstrated that other enzymes may also process the procollagen propeptides. Because of these alternative procollagen I propeptide cleavage mechanisms, the interest in targeting PCP and PNP has diminished [91,92].

Still, scientists are exploring modulating the PCP activity by inhibiting procollagen C proteinase enhancer-1 (PCPE-1, Figure 3). Research has demonstrated that this protein is pivotal in mediating PCP activity in vivo. Further, the biosynthesis of this enhancer is upregulated in fibrotic conditions, including in the skin, heart, liver, kidney, lungs, ligament, muscle, eye, and other tissues and organs [93]. So far, however, no specific PCPE-1 inhibitors have been developed. Controversies exist regarding the safety of inhibiting the activity of this protein, in particular in the context of its roles in broad biological processes, including angiogenesis, cell proliferation, and RNA stabilization [93].

### 7.2. Blocking Collagen Self-Assembly into the Fibrils

One of the newer concepts for reducing the extracellular buildup of collagen I-rich fibrotic deposits is blocking the assembly of collagen molecules into fibrils [64]. Scientists demonstrated that a rationally engineered monoclonal antibody that targets the C-terminal telopeptide of collagen I prevents the aggregation of blocked collagen molecules into fibrils (Figure 3) [94,95]. The anti-fibrotic activities of this anti-collagen antibody (ACA) have been demonstrated in animal models of arthrofibrosis, skin fibrosis, and lung fibrosis [64,96,97,98].

Recently, Steplewski et al. demonstrated that blocking collagen I fibril formation with the antibody accelerates the degradation of ACA-blocked collagen molecules not incorporated into fibrils and speeds up the remodeling of injury sites. Furthermore, detailed analyses of tissues from animals treated with the ACA continuously for two months found that this antibody was safe and caused no side effects [98].

Therefore, the ACA-based method to limit fibrosis offers an attractive anti-fibrotic approach that targets the early onset of the extracellular process of fibril formation. Since collagen fibrillogenesis is a prerequisite for scar formation in all tissues and organs, targeting this process may be a versatile therapeutic approach to limit excessive scarring.

Other inhibitors of collagen fibrillogenesis have also been tested in vitro. Studies demonstrated that (±)-α lipoic acid, trigonelline hydrochloride, oleuropein, capsaicin, soluble discoidin domain receptors (DDR), and fibromodulin block collagen fibrillogenesis by directly interacting with collagen molecules [99,100,101,102,103,104].

Unlike with the ACA, however, the anti-fibrotic utility of these molecules has not been studied in relevant animal models of fibrotic scarring. However, the mechanism of blocking collagen fibrillogenesis with these molecules warrants future tests, to establish their anti-fibrotic potential and safety.

### 7.3. Reducing the Collagen Cross-Linking

Tissue repair via scarring involves scar tissue remodeling. Various enzymes, most notably matrix metalloproteinases (MMPs), degrade the scar elements during this process. Simultaneously, biosynthesis of new ECM elements rebuilds tissue architecture. In optimal conditions, this process transforms a stiff scar into a structure more compatible with the surrounding native tissue.

One of the hurdles that alters the degradation stage of remodeling is the resistance of collagen-rich mature matrices to proteolytic degradation. This resistance is mainly caused by the intrinsic stability of collagen molecules, their tight packing in the fibrils, and covalent cross-links that make fibrillar deposits a formidable target for proteolytic degradation (Figure 3).

In addition to negatively impacting the remodeling of scar tissue, cross-linking stiffens the scar’s ECM, further enhancing its pro-fibrotic characteristics.

Studies have found that those characteristics create mechanical, pro-fibrotic signals not only to fibroblasts, that continue to produce the ECM components, but also to immune cells that participate in healing. Research performed in animal models indicates that a stiff matrix promotes a pro-fibrotic inflammatory response in macrophages and prompts them to produce collagen and other macromolecules that contribute to scar formation [105,106]. Studies have also suggested that a key player that enables the mechanosensing properties of macrophages is an ion channel, PIEZO1 [107].

Because of its involvement in creating a pro-fibrotic matrix environment, collagen cross-linking represents an attractive target to limit excessive scar formation and thus soften the neotissue formed in response to injury.

#### 7.3.1. Inhibiting LOX Activity

One of the indications that blocking LOX-mediated cross-linking reduces fibrotic scarring is an experiment with β-aminopropionitrile (BAPN), that chelates the copper ions needed for proper LOX activity. In animal models, treatment with this compound reduced bleomycin-induced pulmonary fibrosis, CCl_4_-induced liver fibrosis, and esophageal scarring caused by alkali burn [108,109,110].

To create a clinically relevant LOX inhibitor, researchers identified lysyl oxidase-like 2 (LOXL2), a member of the LOX family, as a critical player in the progression of many fibrotic disorders, including pulmonary, cardiac, and tumor-associated fibrosis [111].

Given its pro-fibrotic role, LOXL2 has become a valid anti-fibrotic target (Figure 3). Barry-Hamilton et al. demonstrated that a LOXL2-specific monoclonal antibody inhibits fibrotic changes in cancer, liver, and pulmonary fibrosis models [112]. Subsequently, a humanized IgG4 variant of the anti-LOXL2 antibody (Simtuzumab, Gilead Sciences, Inc., Foster City, CA, USA) was engineered. Its utility to block pulmonary, liver, and tumor-associated fibrosis was tested in clinical trials. These trials, however, were terminated due to the lack of efficacy of simtuzumab [113,114,115,116].

Despite positive outcomes in rodent-based models, it remains unclear why blocking LOXL2 in humans demonstrates no appreciable beneficial anti-fibrotic effects. Puente et al. suggested this could be due to the uninterrupted activity of other collagen cross-linking enzymes, including other members of the LOX family and tissue transglutaminase [117].

#### 7.3.2. Targeting Transglutaminase 2 (TG2)

TG2 is a multifunctional enzyme, able to catalyze the formation of cross-links between the ε-amino group of a lysine residue and a γ-carboxamide group of glutamine residue. TG2-mediated cross-linking of the collagen-rich matrix plays a significant role in fibrosis progression, and blocking this enzyme reduces fibrosis-associated collagen deposition (Figure 3). Oh et al. demonstrated that in TG2-knockout mice, bleomycin-induced pulmonary fibrosis was significantly reduced due to the attenuation of collagen deposition [118]. Similarly, an irreversible TG2 inhibitor reduced fibrosis in a rat model of kidney fibrosis and mouse models of nephrosclerosis, myocardial infarction, and peritoneal fibrosis [119,120,121,122].

Studies have demonstrated that, in addition to stiffening the ECM and making it resistant to proteolysis, TG2 enhances TGF-β1 pro-fibrotic functions. Troilo et al. showed that this enhancement depends on the TG2-mediated multimerization of latent TGF-β binding protein 1 (LTBP1) [123]. The authors suggested that LTBP1 oligomers enhance TGF-β1 binding and activation of this factor by mechanical forces.

Efforts to develop TG2 inhibitors identified a group of molecules that act as competitive amine, reversible allosteric, or irreversible inhibitors [124,125]. These inhibitors, however, lack TG2-blocking specificity, and their use may cause unwanted side effects. Harrison et al. demonstrated that TG inhibitors caused hyperproliferation and parakeratosis in a skin model [124]. Further, Freund et al. demonstrated that targeting TG may inhibit crucial coagulation factor XIIIa [126].

In search of a specific TG2 inhibitor, scientists have developed a therapeutic anti-TG2-antibody and defined its specific binding epitopes. Subsequent studies demonstrated promising results, showing reduced ECM accumulation in a cell-based fibrosis model [127]. Since antibodies are considered effective and safe therapeutics, these results hold promise for developing effective and specific TG2 inhibitors.

## 8. Integrins-TGF-β Activation Axis

Integrins comprising the αv subunit and the β1, β3, β5, β6, or β8 subunit are crucial players in organ fibrosis disorders [128]. Suggested pro-fibrotic mechanisms of action of these integrins include activation of latent members of the TGF-β family.

TGF-β homodimers are synthesized as pro-TGF-β precursors linked covalently with the latency-associated protein (LAP, Figure 5) [129,130]. Following intracellular cleavage of the pro-domain by furins, the mature TGF-β remains non-covalently associated with LAP, forming the small latent complex (SLC). Dissociation of TGF-β from this complex is needed to activate TGFBR type 1 and type 2 [131].

Studies have found that in most cell types, the SLC is secreted in the insoluble form as the large latent complex (LLC), formed intracellularly by the association of TGF-β with the latent TGF-β-binding proteins (LTBPs) [132,133]. LLC interacts with ECM elements following secretion into the extracellular space [134,135,136]. Consequently, ECM-bound LLC is a reservoir of latent TGF-β, which must be released from the complex to activate the TGFBRs.

### 8.1. Integrin-Mediated Enzymatic TGF-β Activation

Two essential mechanisms of TGF-β activation release this factor from the LLC complex. The first mechanism involves enzymatic cleavage of the complex. PCP and various MMPs catalyze proteolysis of the complex [137,138,139]. Studies have suggested that the αv integrins participate in this process by combining the LLC and the LLC-digesting enzymes, facilitating the cleavage and release of active TGF-β. These integrins further optimize the TGF-β functions by enabling the proper spatial arrangement of LLC and TGFBR [140,141].

### 8.2. Integrin-Mediated TGF-β Activation by Cell Traction Forces

TGF-β activation is also achieved by releasing this factor from the LLC without the involvement of proteolytic enzymes [142,143]. In this case, TGF-β is freed from the latent complex by mechanical forces mediated via αv integrins interacting directly with the LAP element of the LLC complex covalently bound to the ECM (Figure 5).

In the stiff ECM environment, the αv-mediated cell binding to LAP transmits cell-generated traction forces in a way that causes LAP deformation and, consequently, TGF-β release. In contrast, cell traction forces are inefficient in deforming LAP in the compliant ECM environment due to the non-resisting matrix. Thus, in the soft ECM, this mechanism of cell traction force-dependent release of TGF-β is less effective than that of the stiff matrix environment (Figure 5).

## 9. Targeting αv Integrins to Reduce TGF-β-Mediated Pro-Fibrotic Cell Behavior

Targeting TGF-β has been recognized as a potent anti-fibrotic strategy [144]. However, it is now clear that direct TGF-β1 targeting is associated with severe side effects [145]. Consequently, scientists have explored the possibility of indirectly blocking this factor’s activity by targeting the various mediators associated with TGF-β signaling. One potential target is CTGF. Various tests have demonstrated that blocking this growth factor reduces fibrotic processes in many disorders [146,147].

Similarly, discovering the role of the αv integrins in TGF-β activation has opened the possibility of blocking this factor via interfering with the activation process. Experimental studies with CWHM12, a synthetic pan-inhibitor that targets all αv integrins, demonstrated attenuation of fibrosis in mouse models of liver, lung, heart, and skeletal muscle fibrosis [148]. Moreover, MK-0429, an αv integrin inhibitor, effectively blocked kidney fibrosis in rats [149].

Integrin-specific blockers have also been developed and tested (Figure 3). One small-molecule compound 8 (c8), that blocks αvβ1 integrin, demonstrated efficacy in renal, liver, and pulmonary fibrosis in mice [150,151]. Further, a cyclic RGD peptide (cilengitide) has shown anti-fibrotic activity by blocking the αvβ3 and αvβ5 integrins in a murine scleroderma model. Similarly, anti-αvβ5 integrin antibody reduced the pro-fibrotic behavior of fibroblasts derived from patients with localized scleroderma [152].

The αvβ6 integrin was also a target of anti-fibrotic treatments. In one example, Patsenker et al. demonstrated that αvβ6 antagonist EMD527040 reduced biliary fibrosis in a murine model [153]. The authors concluded that this reduction was due to the attenuation of TGF-β1 activation, thereby suggesting the utility of this integrin in blocking the progress of a broad range of fibrotic disorders.

Mouse-based studies have demonstrated that αvβ6 integrin-neutralizing antibodies attenuate renal fibrosis in Alport mice [154]. Consequently, the anti-αvβ6 therapeutic antibody, STX-100, or BG00011, was developed for clinical tests. Phase 2 clinical trials with this antibody were conducted to reduce fibrosis in kidney transplants. However, these trials (NCT00878761) were terminated due to unspecified safety concerns [155].

The STX-100 antibody was also applied in clinical trials (NCT03573505) to inhibit IPF and improve FVC parameters. These tests, however, did not show any benefits of the STX-100 antibody. Patients with IPF who received the antibody showed worsening fibrosis compared to the placebo group [156]. Consequently, the trial was terminated.

### Other Integrins as Potential Anti-Fibrotic Targets

Other integrin types may play similar roles in the pathology of fibrotic disorders and should therefore be considered potential anti-fibrotic targets (Figure 3) [157].

Studies have demonstrated that among integrins that recognize the RGD protein motif, including αIIbβ3, α5β1, and α8β1 integrins, the α8β1 integrin is the most attractive target. Blocking this integrin with neutralizing antibodies demonstrated liver and pulmonary fibrosis attenuation in murine models [157,158,159].

Although none of the tested integrin inhibitors have progressed to clinical trials thus far, integrins continue to be an attractive therapeutic target to limit TGF-β-mediated fibrosis [160].

## 10. DDRs as an Anti-Fibrotic Target

DDR1 and DDR2 are collagen-specific receptors that belong to a family of receptor tyrosine kinases (RTKs) (Figure 3) [161,162].

In physiological conditions, these receptors play a pivotal role in embryonic development, growth, wound healing, and tissue homeostasis. They are widely distributed in various tissues. For instance, DDR1 has been detected on epithelial cells in normal tissues and fibrotic areas of the skin, liver, lung, and kidney [163]. Interstitial collagens and collagen types IV and VIII activate this receptor. DDR2 is expressed explicitly on mesenchymal cells and is activated by collagen types II, X, and interstitial collagens [164].

Research data indicate that DDRs are upregulated in fibrotic disorders, including IPF [165]. Studies found that deleting DDR1 reduced fibrosis in adipose tissue in a murine model of cardiometabolic disease. Similarly, knocking out DDR2 attenuated fibrotic changes in renal interstitial fibrosis [166,167]. These results indicate that DDRs participate in pro-fibrotic mechanisms and are valid targets for anti-fibrotic treatments.

As shown by Tao et al., the CQ-061 inhibitor of DDR1 effectively reduced the accumulation of collagen and fibronectin in TGF-β1-activated cultures of human lung fibroblasts [165]. In another example, small-molecule inhibitors of DDR1, imatinib and disatinib, pan-kinase inhibitors, were used as prototypes to develop more specific DDR1 inhibitors [168,169,170].

Although progress has been made in developing DDR inhibitors using existing general RTK blockers as molecular templates, Moll et al. identified some associated challenges [163]. The authors suggested that these challenges occur due to the conserved nature of ATP-binding pockets in all RTKs. As these pockets serve as targets for competitive inhibition of RTKs, blocking the DDR activity in a specific way is difficult. It may therefore be necessary to consider tests of type IV kinase inhibitors that are substrate-competitive rather than ATP-competitive [171,172].

Furthermore, novel drug-screening approaches, such as screening DNA-encoded libraries, may lead to future discoveries of DDR-specific inhibitors. Moll et al. reported some progress in identifying DDR1-specific inhibitors utilizing this screening approach [163].

## 11. Other ECM Anti-Fibrotic Targets

### 11.1. ED-A Fibronectin

Additional potential high-value targets have also been identified, including a fibronectin variant containing the type III extra domain A (ED-A fibronectin). Although ED-A is expressed commonly during non-fibrotic wound healing, it has also been detected in fibrotic lesions [173,174,175,176]. Various studies suggested that ED-A fibronectin activates pro-fibrotic myofibroblasts that produce a stiff ECM, providing a physical environment facilitating the integrin-mediated release of active TGF-β1 [177]. As discussed above, this creates a perfect storm for the acceleration of fibrosis.

Because of its role in fibrosis, the utility of ED-A fibronectin was evaluated in various experimental models. Studies utilizing the ED-A fibronectin function-blocking antibodies or synthetic peptides demonstrated a reduction in TGF-β1 activation and differentiation toward myofibroblasts, indicating that the ED-A variant could serve as a potentially helpful target to limit fibrosis [177,178].

### 11.2. Matricellular Proteins

Matricellular proteins (MCPs) are also considered extracellular anti-fibrotic targets (Figure 3). These proteins belong to a diverse family of matrix molecules that do not contribute directly to building the mechanical structure of the ECM [179]. Although in healthy adult tissues, the expression of MCPs is relatively low, during wound healing, their expression increases significantly [180,181,182].

MCPs fulfill their functions via their ability to modulate the communication between the structural elements of the ECM and cells. They bind to the ECM components and cellular receptors [183]. The receptors that participate in the MCPs’ functions include many integrins, syndecan-4, CD44, endoglin, and others [181].

MCPs have been recognized as crucial players in normal wound healing and fibrosis. Although mechanisms of their involvement are complex and poorly understood, evidence exists for their influence on myofibroblasts’ pro-fibrotic functions. In one example, an increased expression of CCN1 MCP was observed in IPF patients’ myofibroblasts in fibrotic lesions. This increase in CNN1 expression correlated with increased production of fibrotic proteins, including collagen I and fibronectin [182].

One of the critical pro-fibrotic players among MCPs is CTGF, also known as CCN2. This facilitator’s role in TGF-β1 pro-fibrotic functions has been described in many fibrotic disorders, including scleroderma and IPF [184,185].

Accordingly, an anti-CTGF antibody (FG-3019, pamrevlumab) has been selected for clinical tests that target IPF patients. The results of a phase 2 trial demonstrated improvements in the antibody-treated group compared to the placebo control.

Pamrevlumab attenuated the decline in FVC by 60% at week 48 of the treatment. Moreover, the tests indicated that the antibody was well tolerated, and its safety was similar to placebo [186]. At present, pamrevlumab is undergoing a phase 3 clinical trial (NCT03955146).

This encouraging example of CTGF targeting in patients with IPF suggests that MCPs may be attractive targets to limit the progress of fibrosis in other tissues and organs. Therefore, the utility of blocking CTGF in other fibrotic conditions has also been analyzed. Barbe et al. demonstrated a reduction in skeletal muscle fibrosis in rats treated with pamrevlumab [187]. Similarly, Vainio et al. utilized this antibody to attenuate fibrosis in myocardial infarction and improve the repair of the cardiac muscle in a murine model [188].

Other MCPs, including osteopontin (OPN), also play pro-fibrotic roles. While in healthy adult tissues the OPN expression is relatively low, in fibrotic conditions, the expression of this protein increases significantly. This increase is associated with myofibroblasts activation and increased collagen deposition [189,190,191,192].

Clinical and experimental data suggest the pro-fibrotic function of OPN in cardiovascular diseases with fibrotic features, including dilated cardiomyopathy and post-myocardial infarction injuries [193,194,195,196]. The role of this protein in cardiac fibrosis was strongly supported by various studies of cardiac fibrosis performed in OPN-null mice. These studies demonstrated that, in contrast to the wild-type mice, the mice lacking OPN had a significantly attenuated fibrotic response to pro-fibrotic stimuli [197,198,199].

Subsequently, OPN was included in the list of anti-fibrotic targets that blocked MCP-associated functions. In one study, Dai et al. showed that the OPN-neutralizing antibodies attenuated functional decline in heart functions in a murine model of dilated cardiomyopathy [200].

The above data suggest that MCPs may remain an attractive anti-fibrotic target worth exploiting beyond the MCP candidates identified thus far.

### 11.3. Targeting the Extracellular Vesicles in Organ Fibrosis

Extracellular vesicles (EVs) have recently been identified as targets for limiting fibrosis (Figure 3). These structures include exosomes, microvesicles, and apoptotic bodies, each characterized by distinct formation mechanisms and specific cargo [201].

EV formation includes the budding of cellular membranes via evagination or invagination. These budding mechanisms form vesicles that envelope various materials, including proteins, lipids, mRNA, and micro(mi)RNA. As EVs are released to the extracellular space, they occur in bodily fluids and the ECM.

The EVs modulate the behavior of cells by fusing with them and releasing specific cargo. In fibrotic conditions, EVs may carry pathological products formed due to injury, inflammation, and pro-fibrotic cell activation.

Some have suggested that EVs may help to develop practical anti-fibrotic approaches due to their properties of cell homing and the ability to reprogram the behavior of target cells [202]. Lenzini et al. suggested that, due to the aquaporin-1-dependent regulation of EV hydration, these vesicles are uniquely suited to penetrate a dense fibrotic tissue structure, facilitating efficient cargo delivery [203].

In support of the EVs’ involvement in fibrosis, research has demonstrated their increase in many fibrotic organs, including the lung, kidney, heart, pancreas, skin, and others [204,205,206,207,208]. Consequently, various research groups have explored the possibility of utilizing EVs to attenuate fibrotic responses to organ and tissue injuries. They proposed that applying EVs from non-fibrotic sources to fibrotic tissues and organs would provide therapeutic effects.

Many preclinical studies have indicated the potential utility of EV-based therapies. In one study, exosomes extracted from human bone marrow mesenchymal stem cells (BM-MSC) prevented and reversed pulmonary fibrosis in mice treated with bleomycin [209]. The authors demonstrated that these positive outcomes were due to changes in the macrophage population, that switched their phenotype from pro-inflammatory to homeostatic. Similarly, utilizing a hyperoxia-induced bronchopulmonary dysplasia model, Wills et al. demonstrated that applying exosomes isolated from human MSCs improved lung function via macrophage-associated mechanisms [210].

Other cell sources of the EVs suitable for reducing pulmonary fibrosis tested in animal models included amnion epithelial cells. Exosomes isolated from these cells attenuated the inflammation, epithelial damage, deposition of fibrotic ECM, and expression of TGF-β and reduced the number of myofibroblasts [211,212].

Other sources of EVs tested to improve fibrotic lung functions included macrophages [213]. Studies have established that exosomes isolated from macrophages suppress the biosynthesis of TGFBRs and collagen I. One study documented that this reduction was enabled by mechanisms involving anti-fibrotic miRNA-142-3p present in the macrophage-derived exosomes.

Similar therapeutic approaches utilizing EVs were evaluated in other organ fibrosis models [201]. Wang et al. employed human BM-MSC-derived EVs loaded with anti-fibrotic miRNA-101a. They demonstrated their protective functions in a mouse model of myocardial infarction. In particular, they observed the miRNA-101a-mediated reduction in pro-fibrotic TGF-β1, TGF-β2, and collagen [214].

In a comprehensive review, Brigstock presented further details on the clinical utility of EVs derived from various sources [201]. The author pointed out the significant potential of EVs to play a positive role in anti-fibrotic approaches. He also emphasized that our understanding of EVs’ functions in biology and pathology, including fibrosis, is still in its infancy. One factor limiting our understanding of these functions in vivo is that most of the data generated thus far are derived from cell-based studies. Therefore, more biologically relevant preclinical studies are needed to fully comprehend the prospects and limitations of EV-based approaches to reduce fibrosis.

## 12. Conclusions

Localized and systemic fibrotic disorders continue to result in severe medical illness and, thus, social burden. Despite substantial scientific efforts to mitigate fibrotic disease, few fibrosis-specific therapeutics have been approved for limited clinical use.

Although various tissues and organs have distinct biological functions and molecular and cellular compositions, they equally respond to fibrotic stimuli by synthesizing collagen-rich scars. Consequently, universal anti-fibrotic targets must be defined against shared pro-fibrotic mechanisms to develop broad-use anti-fibrotic therapeutics. In this context, extracellular targets, potentially limiting fibrotic healing in response to tissue injury, have become a focal point of many anti-fibrotic approaches. The central premise of these approaches, is that by modulating crucial elements of mechanisms that propagate the formation of the pro-fibrotic stiff matrix, it may be possible to reduce excessive scarring. Several potential targets associated with matrix stiffening, including LOX, TG2, PCP, and others, were identified. Their therapeutic utility has been tested at both the preclinical and clinical levels. However, despite promising preliminary results, most of these targets failed to meet the required expectations to be considered therapeutically valid.

Nevertheless, because of the essential role of the ECM in the structure and function of distinct tissues and organs during excessive scar production, targeting the ECM remains an attractive strategy to limit fibrosis. Such an approach offers the possibility of developing therapeutics to treat various fibrotic disorders, regardless of the injury site or location.

We propose that to improve the outcomes of studies targeting extracellular scarring mechanisms, it will be necessary to (i) employ relevant animal models of fibrotic disorders to test anti-fibrotic approaches in biologically relevant conditions, (ii) apply stringent criteria to describe the outcomes of anti-fibrotic approaches at the molecular, cellular, and tissue levels simultaneously, (iii) target early stages of stiff matrix formation, (iv) aim concomitantly at multiple targets, and (v) focus on mitigating excessive fibrosis rather than resolving established fibrotic tissue.

## Figures and Tables

**Figure 1 biomolecules-13-00758-f001:**
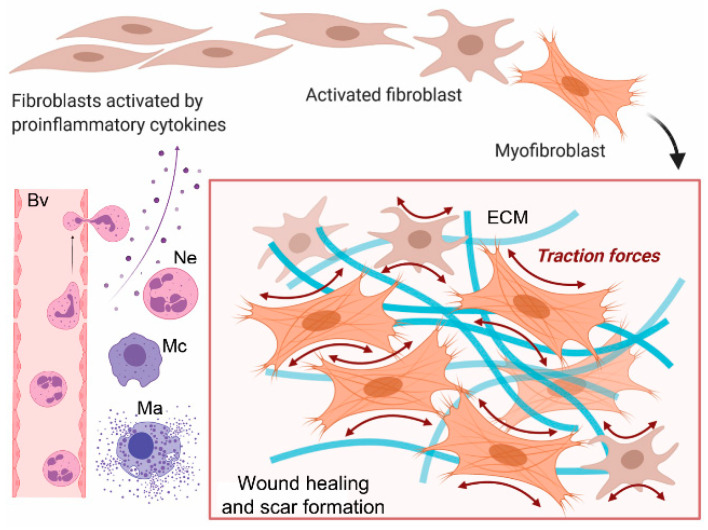
Crucial steps in fibroblast activation. A blood vessel (Bv), neutrophils (Ne), macrophages (Mc), and mast cells (Ma) are indicated.

**Figure 2 biomolecules-13-00758-f002:**
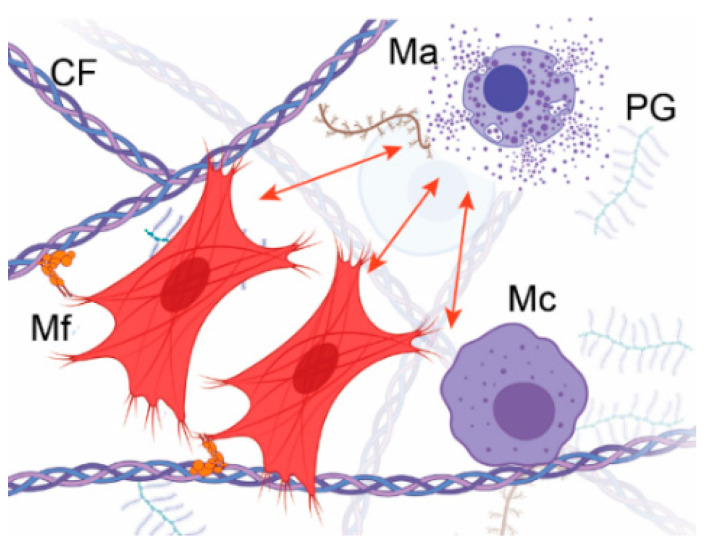
A schematic showing activation of mast cells (Ma) and macrophages (Mc) by forces generated by myofibroblasts (Mf) embedded in the ECM and acting via integrins. Collagen fibrils (CF) and proteoglycans (PG) are indicated.

**Figure 3 biomolecules-13-00758-f003:**
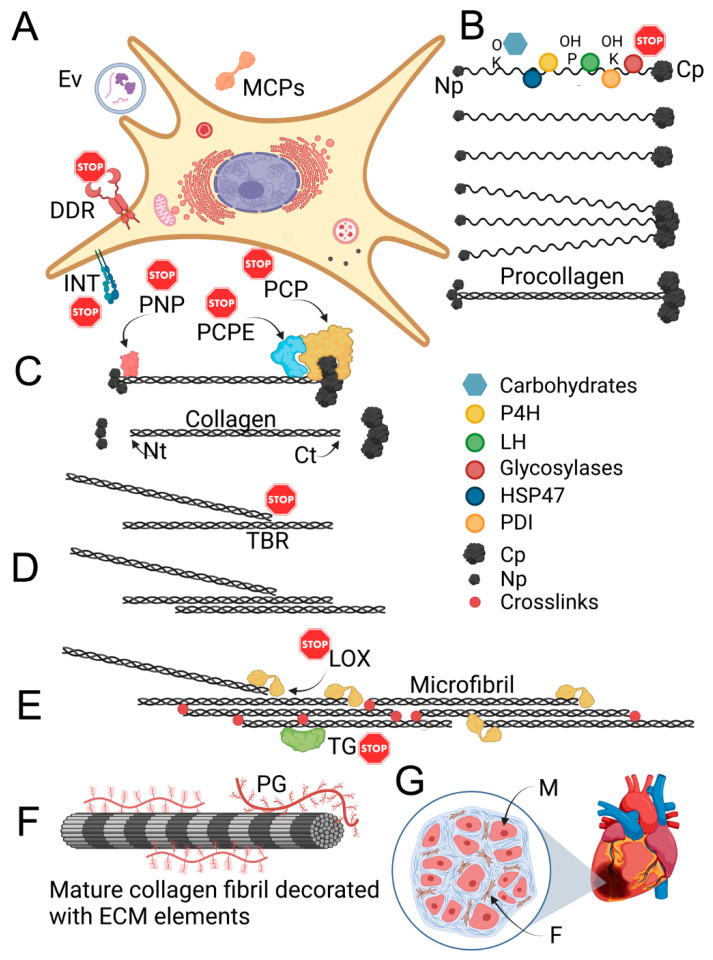
A schematic depicting crucial elements of mechanisms that contribute to the formation of collagen-rich, stiff matrices that promote excessive scarring. The STOP signs indicate the elements whose blocking is associated with anti-fibrotic effects. (**A**) A fibroblastic cell that includes discoidin domain receptors (DDR) and integrins (INT). Matricellular proteins (MCPs) and extracellular vesicles (Ev) are also indicated. (**B**) Intracellular biosynthesis of procollagen molecules formed by trimerization of individual procollagen chains. Crucial collagen-modifying enzymes include prolyl 4-hydroxylase (P4H), lysyl hydroxylase (LH), and glycosylases. Moreover, heat-shock protein 47 (HSP47) and protein disulfide isomerase (PDI) exemplify protein chaperones participating in procollagen formation. (**C**) Extracellular processing of procollagen propeptides with procollagen N proteinase (PNP) and procollagen C proteinase (PCP), whose activity is accelerated by PCP enhancer (PCPE). The N-terminal (Np) and the C-terminal (Cp) propeptides are also indicated. The N-terminal (Nt) and the C-terminal (Ct) telopeptides are also indicated. (**D**) Site-specific self-assembly of collagen molecules into a fibril; the assembly is driven by the interaction of collagen telopeptides with an interacting partner’s telopeptide-binding region (TBR). (**E**) A depiction of a collagen microfibril, in which collagen molecules undergo cross-linking catalyzed by lysyl oxidases (LOX) and transglutaminases (TG). (**F**) A mature collagen fibril associated with other structural macromolecules, e.g., proteoglycans (PG). (**G**) An example of a fibrotic site that affects an injured organ. In addition to muscle cells (M), a magnified insert shows fibroblasts (F) embedded in collagen-rich fibrotic tissue.

**Figure 4 biomolecules-13-00758-f004:**
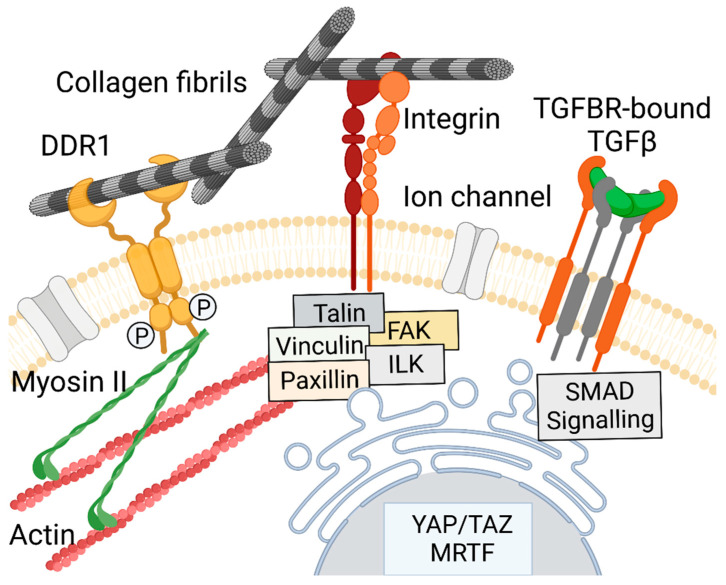
A diagram showing crucial players involved in mechanical signal transduction in fibrotic scars. Collagen fibril-activated and phosphorylated discoidin domain receptor 1 (DDR1), in association with myosin II, is indicated. A matrix-bound integrin with its cytoplasmic interactants, comprising talin, vinculin, paxillin, focal adhesion kinase (FAK), and integrin-linked kinase (ILK), is also presented. In addition, a TGFBR bound to its TGF-β ligand is demonstrated. The diagram also illustrates a nuclear location of yes-associated protein (YAP), transcriptional co-activator with PDZ-binding motif (TAZ), and myocardin-related transcription factor (MRTF).

**Figure 5 biomolecules-13-00758-f005:**
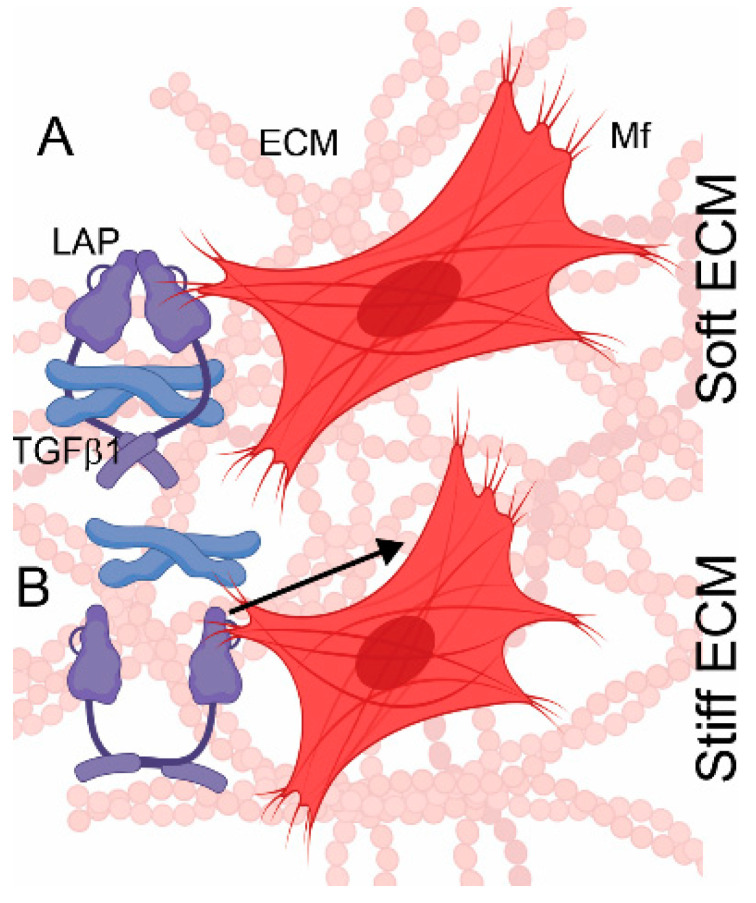
A depiction of TGF-β1 activation by the mechanical forces (arrow) generated by myofibroblasts (Mf) embedded in soft ECM (**A**) or stiff ECM (**B**). TGF-β1 and latency-associated protein (LAP) are indicated.

## Data Availability

Not applicable.

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
