# Peer review of "Extracellular Targets to Reduce Excessive Scarring in Response to Tissue Injury"

_biomolecules, 2023, doi:10.3390/biom13050758_

Round 1

Reviewer 1 Report

Dear Authors,

the article „Extracellular targets to reduce excessive scarring in response to tissue injury” authored by Fertala et al. provides a comprehensive overview and the rationale for studies targeting the extracellular aspects of fibrosis, and discusses the progress and limitations of current extracellular approaches to limit fibrotic healing. This is a well-written, comprehensive paper quoting 213 papers, divided into 12 chronologically arranged thematic sections, including introduction and conclusions, and 5 clearly presented figures.

Excessive fibrosis leading to scar formation results in organs failure, and ultimately death. It is estimated that fibrotic disorders are associated with 45% of all deaths. Additionally, most of anti-fibrotic treatments, including also those under experimental stage seem to be largely unsuccessful. Considering that fibrotic disorders constitute a serious medical problem worldwide, and there is still an important need to find out relevant targets to counteract fibrosis, the topic discussed by Fertala et al. seems to be of high priority for publication.

Nevertheless, there are some minor suggestions that need to be included in the manuscript before its publication:

  1. Page 3, Line 78: the part of sentence: „significant scarring can severely alter the repaired tissue function” should be replaced by: “significant scarring can severely alter the repaired tissue architecture and function”
  2. Page 6, Figure 3: This schema is focused on the intra- and extracellular mechanisms of collagen production in the stiff matrices promoting excessive scarring, therefore the presence of PG molecule in panel B is not necessary and should be removed. PG presence in panel F together with its mention in the figure 3 legend is satisfactory
  3. Page 8, line 306: “endothelial-to-mesenchymal transition” should be corrected to “epithelial-to-mesenchymal transition”
  4. Page 8, Figure 4, legend title: “A diagram showing crucial players involved in mechanotransduction” would seem better if changed as follows: “A diagram showing crucial players involved in mechanical signals transduction in the fibrotic scars”
  5. Page 10, lines 412-414: “Research done in animal models indicates that stiff matrix promotes macrophages to behave in a pro-fibrotic way and produce collagen and other macromolecules that contribute to scar formation [104,105]” – this sentence is not clear and needs to be explained how “macrophages behave in a pro-fibrotic way”? Is it because macrophages itself start to produce collagen or whether collagen production is upregulated in stiff matrix-derived fibroblasts due to regulation by macrophages?

Author Response

We thank the reviewers for their supportive comments about our review article and their constructive suggestions on how to improve it. In response, we amended our manuscript to accommodate the reviewers’ comments. The following paragraphs summarize our responses:

Reviewer 1.

            This reviewer suggested making specific changes to text lines 78, 306, and 412-414. In response we included all suggestions.

            Moreover, this reviewer recommended changes to Figure 3 and Figure 4. In response, we removed a diagram representing proteoglycans in panel B (Figure 3) and amended the Figure 4 legend.

As suggested in the reviewer’s point #5, we amended a fragment describing the role of macrophages in fibrosis.

Reviewer 2.

            This reviewer pointed out our omitting glycosaminoglycans and proteoglycans in our review of anti-fibrotic targets. We strongly agree with this reviewer that these molecules play a significant role in wound healing and fibrosis. Although initially we contemplated including these macromolecules, ultimately, we did not.

            Our concern was that because these molecules form an extremely large and diverse family in which members play unique physiological and pathological roles in a tissue-dependent fashion, including them would be beyond the main scope of our review. As suggested by this reviewer, however, we included these molecules in our introduction and provided a relevant reference that describes their role in the physiology and pathology of wound healing.

Best regards,

Andrzej Fertala, Ph.D.

Reviewer 2 Report

The review article authored by Fertala et al. offers valuable insights into the biologically and medically important topic of fibrosis, elucidating the underlying mechanisms and proposing several targets for therapeutic intervention. I found the review to be highly relevant, meticulously researched, and well-organized, reflecting the authors' expertise in the field.

While the manuscript is well-structured, the introductory section may benefit from a more scientific approach. Nevertheless, it provides a useful overview of the topic and can serve as a suitable primer for readers less familiar with the subject matter.

However, I would like to draw attention to an area that could be further elaborated in the manuscript. The authors did not mention GAGs, such as hyaluronan and sulfated proteoglycans, which are important non-proteinaceous components of the ECM that drive tissue architecture and stiffness and bind a large number of growth factors including TGFbeta. These GAGs are known to play a significant role in fibrosis, and their exclusion is notable. Therefore, I suggest that the authors consider including at least a brief mention of GAGs in the introduction or in Chapter 11, where other ECM anti-fibrotic targets are discussed.

Overall, I believe that the review by Fertala et al. is a well-written and informative contribution to the literature on fibrosis, and I recommend it for publication (after minor revision) in the Biomolecules journal.

Author Response

(The authors gave the same response as above.)
